# Human intraparietal sulcal morphology relates to individual differences in language and memory performance
Federica Santacroce [1] ✉, Arnaud Cachia[2,3], Agustina Fragueiro[1], Eleonora Grande[1], Margot Roell [2], Antonello Baldassarre[1], Carlo Sestieri[1] & Giorgia Committeri [1] ✉

The sulco-gyral pattern is a qualitative feature of the cortical anatomy that is determined in utero, stable throughout lifespan and linked to brain function. The intraparietal sulcus (IPS) is a nodal associative brain area, but the relation between its morphology and cognition is largely unknown. By labelling the left and right IPS of 390 healthy participants into two patterns, according to the presence or absence of a sulcus interruption, here we demonstrate a strong association between the morphology of the right IPS and performance on memory and language tasks. We interpret the results as a morphological advantage of a sulcus interruption, probably due to the underlying white matter organization. The right-hemisphere specificity of this effect emphasizes the neurodevelopmental and plastic role of sulcus morphology in cognition prior to lateralisation processes. The results highlight a promising area of investigation on the relationship between cognitive performance, sulco-gyral pattern and white matter bundles.

During foetal brain development, the macroscopic aspect of the cortex changes from a smooth, lissencephalic brain to a folded, gyrencephalic brain[1]. Cortical folding appears in a specific temporal sequence in utero and its spatial organisation/topology, termed the sulco-gyral pattern, is established at birth[2,3]. Unlike other quantitative features of cortical anatomy, such as cortical thickness or volume, which show plasticity with age, learning, or brain damage, the qualitative sulcal pattern remains stable across the lifespan[4]. Accordingly, the study of the qualitative sulco-gyral patterns of the cortex may open a new research perspective and allow testing whether trait markers of the brain anatomy, such as the sulcal pattern, represent fingerprints of inter-individual variability in cognitive performance[5].

Genetic factors have been early proposed in the cortical folding debate (e.g., ref. 6), with particular attention to the rapid increase in the number of neurons putting pressure on the limited space inside the skull, the combination of radial and tangential forces (due to the migration of neurons and their variation in growth rate, respectively), and the variation in density, orientation and connections of white matter fibers (see refs. 5,7 for reviews). Moreover, a relationship between cortical folding and the underlying cytoarchitecture has been suggested[8–10].

Several studies have investigated the relationship between sulcal pattern and cognition in both normal (e.g., ref. 11) and clinical (e.g., ref. 12) populations, focusing on a single cognitive domain. For instance, the anterior cingulate cortex (ACC), with either a "single" or "double parallel" type[13], has been repeatedly investigated in relation to cognitive control[14–16]. In another cognitive domain, Cachia et al. [17] have found a positive association between an interrupted, compared to a continuous, occipito-temporal sulcus (OTS) and reading performance. More recently, a sectioned horizontal intraparietal sulcus (hIPS) has been associated with better math fluency and symbolic number abilities[18]. The authors suggested that the advantage of an interrupted/sectioned pattern might be, at least in part, related to the higher number, greater fiber diameter and/or greater myelination of the underlying white matter.

All these studies have investigated the morphological pattern of cortical areas selected a priori in relation to a single specific cognitive function, based on the hypothesis of "one-to-one mapping"[19]. However, several cortical areas are involved in multiple cognitive functions. For instance, the intraparietal sulcus (IPS), located between sensorimotor and visual areas in the anterior-posterior axis and between the superior and inferior parietal lobules (SPL, IPL) in the dorsal-ventral axis, is considered a nodal associative/integrative brain area also for the many underlying connections[20]. Classical electrophysiological studies in monkeys have reported the presence of functionally specialised IPS regions (e.g., AIP, MIP, LIP) whose

[1]Department of Neuroscience, Imaging and Clinical Sciences, and ITAB, Gabriele d'Annunzio University, Via Luigi Polacchi 11, 66100 Chieti, Italy. [2]Université Paris Cité, Laboratoire de Psychologie du développement et de l'Education de l'Enfant (LaPsyDÉ), CNRS UMR 8240, Paris, France. [3]Université Paris Cité, Institut de Psychiatrie et Neurosciences de Paris (IPNP), INSERM, UMR S1266, Paris, France. ✉e-mail: federica.santacroce@unich.it; giorgia.committeri@unich.it

neural activity is associated with fixation and manipulation of object features, multimodal stimuli, reaching movements and saccades (for review see ref. 21). Subsequent human neuroimaging studies have further demonstrated the involvement of the IPS in higher-level cognitive functions, such as visuospatial attention[22] and motor intention[23], calculation[24], memory[25] and language[26].

In this context, the present study aims to (1) identify the morphological pattern distribution of the hIPS in a large sample of healthy participants; (2) investigate the brain-behavioural relationship between hIPS morphology and performance on multiple cognitive abilities using a data-driven approach. To this aim, we took advantage of the Human Connectome Project, which provides a large dataset of behavioural and neuroimaging measures and selected a sample with a limited age range (26–30 years old) to overcome potential age-related confounds. The cognitive data were subjected to a principal component analysis (PCA) to extract factors of common cognitive variance and reduce the collinearity between the different variables. Using individual structural MRI data, we identified the horizontal branch of the IPS (hIPS) in each participant, classified its pattern as "Interrupted" or "Continuous", and compared the relative frequency of the two anatomical variants with previous IPS morphological studies conducted on smaller cohorts[18,27,28]. Then, we analysed the association between hIPS morphology and the PCA factors capturing behavioural variance, with the hypothesis that an interruption of the hIPS would be advantageous for cognitive functions in addition to numerical cognition, as previously described[18].

Our findings revealed a prevalence of the "interrupted" pattern in the right hemisphere, which was associated with higher behavioural performance in memory and language tasks, putatively due to intra-lobar parietal white matter.

## Results

### Sulcal pattern distribution
Our sample consisted of 390 participants (173 males, 40% - age range 26–30) derived from the Human Connectome Project (HCP). For both hemispheres of each participant, we manually labelled the intraparietal sulcus (IPS) on the inflated and pial cortical surfaces and classified them into two morphological patterns according to the presence or absence of sulcus interruptions along the horizontal branch of the IPS (hIPS) (see Fig. 1). A total of 780 manual labels were created. The criteria for the classification are described in more detail in the "Methods" section. The interrupted hIPS was found in 47.44% of the left hemispheres and in 69.23% of the right hemispheres (see Fig. 2).

The binomial test showed a significant difference between the number of interrupted and continuous hIPS in the right hemisphere ($p < 0.001$, 95% CI of 64.4%–73.8%), whereas no significant difference was found in the left hemisphere ($p = 0.336$, 95% CI of 47.5%–57.6%). Logistic regression

showed that both sex and hemisphere significantly influenced the presence of an hIPS interruption. Specifically, males exhibited a significantly higher presence of interrupted hIPS compared to females ($\chi^2 = 25.263$; df = 1; $p = 5.0 \times 10^{-7}$) and the right hIPS was significantly more likely to be interrupted than the left hIPS ($\chi^2 = 7.329$; df = 1; $p = 0.007$). The overall model was statistically significant ($\chi^2 = 65.079$, $p = 3.6 \times 10^{-14}$), explained 11% (Nagelkerke $R^2$) of the variance and correctly classified 62.8% of the cases.

We then compared the distribution of the hIPS sulcal pattern observed in the present study with that of the three previous studies that investigated the hIPS sulcal morphology[18,27,28]. The $\chi^2$ analysis, controlling for the sample size, showed a significant difference between the four studies regarding the distribution of the left hIPS sulcal pattern ($\chi^2 = 14.1$; df = 3; $p = 0.003$). This difference is due to the predominance of the continuous pattern in Ono et al.[27], of the interrupted pattern in Zlatkina & Petrides[28], and an equal presence of both patterns in Roell et al.[18] and the present study. On the contrary, no significant difference was found in the right hIPS sulcal distribution between the four studies ($\chi^2 = 1.7671$; $p = 0.622$), with a prevalence of the interrupted pattern in all of them (see Fig. 3).

### Principal component analysis on cognitive measures
To examine the relationship between hIPS sulcal pattern and cognition, we performed a Principal Component Analysis (PCA) on several behavioural measures indexing different cognitive functions (see details in "Methods" section). The PCA identified five principal factors with an eigenvalue >1, accounting for 55% of the total variance. The first factor (20.6% of the variance) showed positive saturation with the following tasks: episodic memory (verbal and non-verbal), working memory and language (production and comprehension). Factor 1 was therefore labelled 'Memory and Language'. The second factor (12.1% of the variance) showed negative saturation with the executive functions (inhibition and cognitive flexibility) and speed processing tasks, so it was labelled "Executive Functions". Negative saturation means that high factor scores correspond to poorer performance and vice versa. Thus, for clarity and consistency across factors, we have inverted the corresponding bars in Figure 4 (see below). The third factor (8.9% of the variance) showed positive saturation with delay discounting tasks, so it was labelled "Impulsivity". The fourth factor (6.8% of the variance) showed positive saturation with sustained attention task, so it was labelled "Attention". Finally, the fifth factor (6.5% of the variance) showed positive saturation with intelligence and spatial orientation tasks, so it was labelled "Intelligence and Spatial Orientation". All factors, with their respective tasks and saturation levels, are shown in Table 1.

### Relation between hIPS morphology and cognition
The relationship between morphological features of the hIPS and the five PCA-derived factors was assessed with a Generalized Linear Model (GLM),

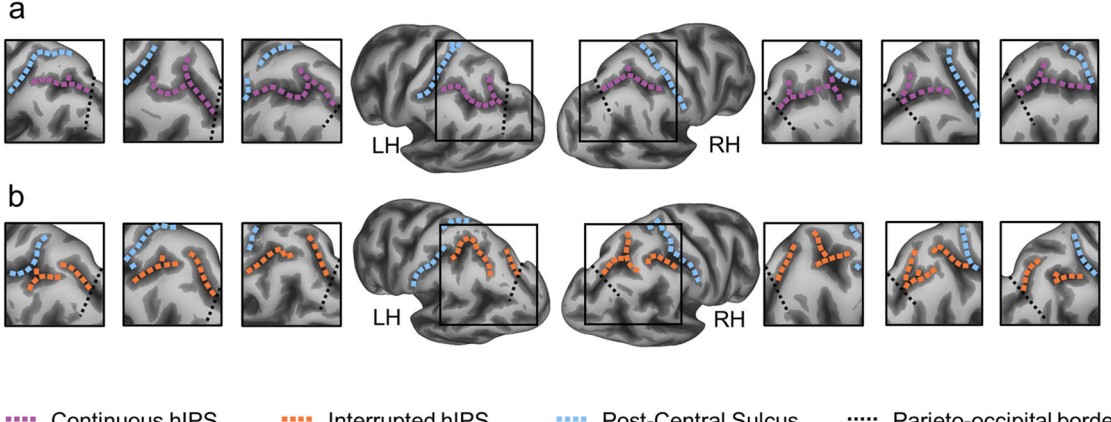

···· Continuous hIPS ···· Interrupted hIPS ···· Post-Central Sulcus ···· Parieto-occipital border

**Fig. 1 | Sulcal patterns of the horizontal intraparietal sulcus (hIPS). a** Exemplars of continuous patterns (sulci shown with violet dashed lines). **b** Exemplars of interrupted patterns (sulci shown with orange dashed lines). LH left hemisphere, RH right hemisphere.

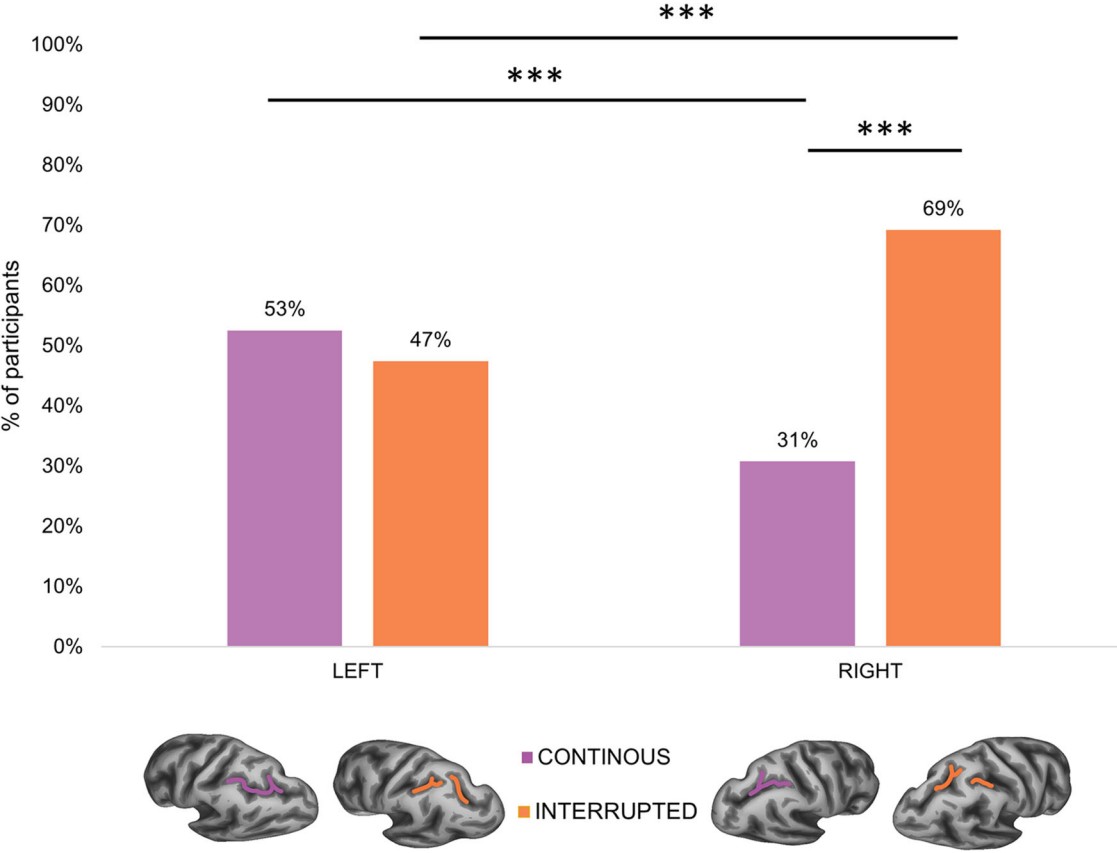

**Fig. 2 | Sample distribution of hIPS sulcal patterns.** Percentage of participants with continuous (in violet) and interrupted (in orange) patterns in both left and right hemispheres. ***$p < 0.001$; sample size $n = 390$.

**Fig. 3 | Comparison with previous studies.** Comparison between the present and previous studies of IPS sulcal pattern sample distribution. Violet lines represent the "continuous" pattern and orange lines the "interrupted" pattern in the left (LH) and right (RH) hemispheres.

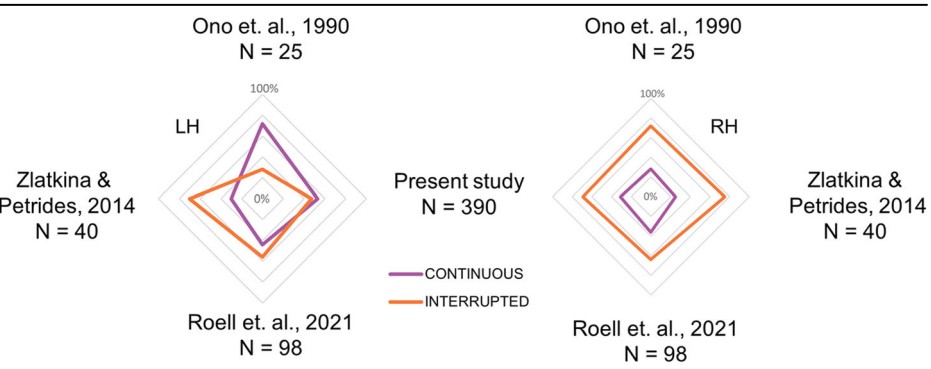

testing separately for the effect of left and right IPS interruption. Specifically, we entered the five behavioural factors as dependent variables and three fixed factors: morphological pattern of the left hIPS ("interrupted" vs. "continuous"), morphological pattern of the right hIPS ("interrupted" vs. "continuous"), and sex ("Males" vs "Females"). This analysis revealed a main effect of the right hIPS pattern ($F = 13.176$; df = 1; $p = 0.0003$; observed power = 0.95) and sex ($F = 4.328$; df = 1; $p = 0.038$; observed power = 0.55) on the first behavioural factor (Memory and Language), whereas the left hIPS pattern had no significant effect ($p = 0.585$). In particular, the right hIPS interruption was associated with better performance at memory and language tasks, and females performed better than males. A significant main effect of sex was also observed for the factors 'Executive Functions' ($F = 6.845$; df = 1; $p = 0.009$; observed power = 0.74) and 'Intelligence and Spatial Orientation' ($F = 9.467$; df = 1; $p = 0.002$; observed power = 0.87). In particular, females performed better than males on 'Executive Functions' and males performed better than females on 'Intelligence and Spatial

Orientation'. No other significant main or interaction effects were found (all $p > 0.05$). Figure 4 shows the mean performance for each cognitive/behavioural factor in relation to the hIPS morphological pattern of both hemispheres.

**Effect of cortical thickness**

Finally, we conducted a control analysis to test whether the cortical thickness, i.e., a quantitative structural feature of the superior and inferior parietal lobules, contributed to the observed significant relationships between the behavioural performance and the right hIPS sulcal pattern, i.e., a qualitative structural feature. The thickness of the left and right Superior Parietal Lobule (SPL) and Inferior Parietal Lobule (IPL) were included as covariates in the statistical model, with the left hIPS morphological pattern, the right hIPS morphological pattern and the sex as independent variables, and the significant first PCA factor as dependent variable. The results showed no significant effect of the right SPL ($p = 0.299$) and IPL ($p = 0.262$) thickness

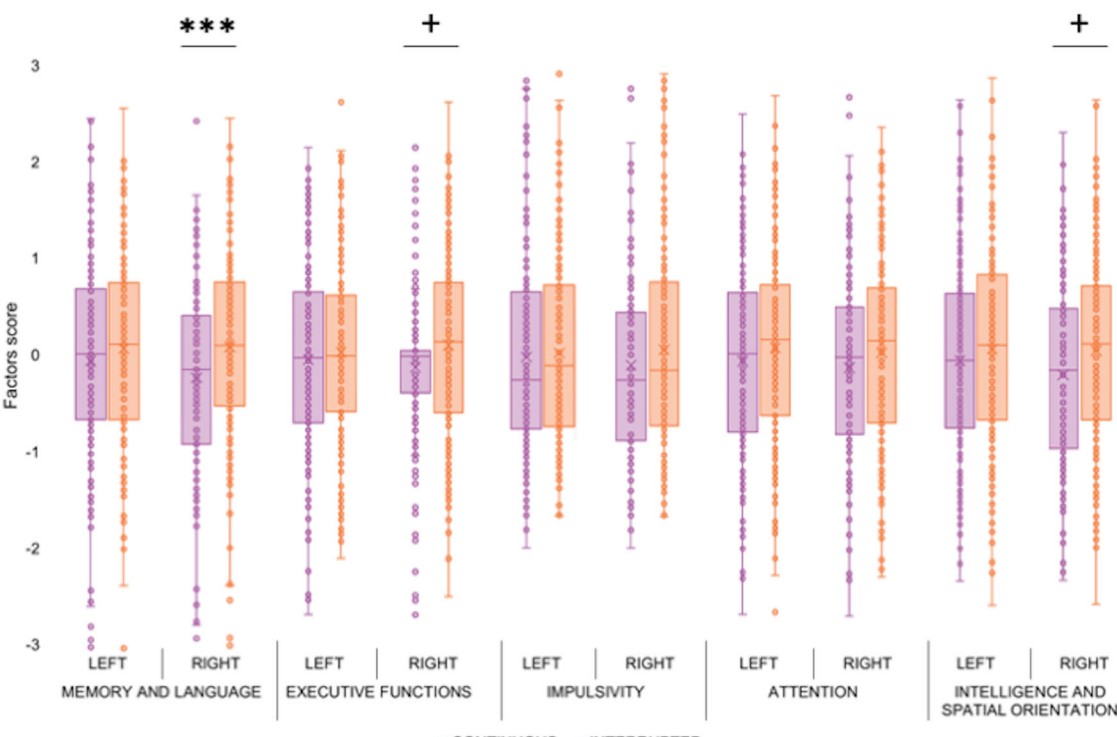

**Fig. 4 | hIPS sulcal pattern and cognition.** Scores of the five principal factors derived from principal component analysis (PCA) in participants with "continuous" (violet dots and bars) and "interrupted" (orange dots and bars) hIPS in the left and right hemispheres. Error bars represent the standard error of the mean; ***$p < 0.001$, +$p < 0.1$; sample size $n = 390$.

**Table 1 | Cognitive factors derived from the principal component analysis (PCA) and corresponding tasks**

| Factor 1: Memory and Language (20.6%) | Factor 2: Executive functions (12.1%) | Factor 3: Impulsivity (8.9%) | Factor 4: Attention (6.8%) | Factor 5: Intelligence and spatial orientation (6.5%) |
|---|---|---|---|---|
| Verbal episodic memory (0.664) | Executive function (inhibition) (−0.862) | Delay discounting (reward 200$) (0.897) | Sustained attention (RT) (0.654) | Intelligence (RT) (0.868) |
| Verbal episodic memory (RT) (−0.577) | Executive Function (cognitive flexibility) (−0.834) | Delay discounting (reward 40 K$) (0.888) | Sustained attention (specificity) (0.612) | Intelligence (0.740) |
| Language (production) (0.555) | Speed processing (−0.544) | | Sustained attention (d′) (0.453) | Spatial orientation (RT) (0.688) |
| Language (comprehension) (0.550) | | | | Spatial orientation (0.401) |
| Working Memory (0.516) | | | | |
| Non-verbal episodic memory (0.448) | | | | |

The labels (first row in bold) represent the five PCA factors. The following rows contain all the cognitive tasks, with a saturation level >0.4, included in each factor (respective saturation level).

and left SPL ($p = 0.723$) and IPL ($p = 0.592$) thickness on the cognitive factor of Memory and Language. Of note, the previously reported effects of qualitative morphological features (right hIPS pattern: $p = 0.0003$) and sex ($p = 0.026$) remained significant also when controlling for quantitative structural features.

## Discussion

The present study reports, for the first time, the relation between the sulcal pattern of the horizontal intraparietal sulcus (hIPS) and several cognitive functions in a large sample of healthy young adults. The hIPS is a critical meeting point of multiple and diverse functional regions, located in a central position in the parietal lobe, close to the nodal angular gyrus hub and its important number of connections.

Consistent with previous studies[18,27,28], the interrupted pattern is more present than the continuous pattern in the right hemisphere. On the contrary, in the left hemisphere, the two patterns are more balanced in our study (53% of continuous pattern and 47% of interrupted pattern) and in Roell et al.[18] but unbalanced towards the continuous pattern in Ono et al.[27] and towards the interrupted" pattern in Zlatkina & Petrides[28]. In this regard, it is important to note the large sample size of the current study ($N = 390$) compared to previous studies. Ono et al.[27] examined only 25 post-mortem brains, Zlatkina & Petrides[28] analysed MRI from 40 participants and Roell et al.[18] examined the effect of the hIPS interruption in 98 participants. Overall, these studies differ in several methodological aspects (sample size, brain images, interruption detection, labelling criteria) and this might have influenced the inconsistent morphological

pattern distributions in the left hemisphere. In particular, the study which reported a significantly higher proportion of interruptions in the left hemisphere[28] (see Fig. 3) is the only one considering the "plis de passage" (also known as submerged gyri or gyral/cortical bridges, see ref. 29) as sulcal interruptions, and finding them as more frequent in the left hemisphere[28].

Regarding the association between the hIPS sulcal pattern and cognition, we found a strong relation of the right hIPS pattern with the performance in memory and language tasks. Specifically, a right hIPS interruption was associated with higher scores on the first PCA factor, which includes episodic memory (verbal and non-verbal), working memory and language (production and comprehension) tasks. Consistent with previous evidence[17,18,30], sulcal interruption appears to be advantageous and therefore to represent a positive, early defined, fingerprint for cognitive development. The presence of a sulcal interruption may reflect an increase in the underlying microstructure of both the white matter bundles and the cortical ribbon[7]. Therefore, it might represent an indirect marker of an increased amount of cortical tissue and connections available for neuroplastic processes underlying the acquisition of cognitive functions. Notably, the association of hIPS structure with memory and language appears to be specific to the sulcal morphology of this region, as control analyses showed no effect of the cortical thickness in this region.

Recent findings provide evidence for the involvement of the IPS in complex cognitive functions such as episodic memory[25,31], language[32,33] and working memory[34]. This set of high-level cognitive functions may share a common core substrate in the inferior parietal lobule (IPL), related to its direct auditory and visual inputs from the temporal and occipital cortex[35] and to somatosensory and proprioceptive information from the postcentral gyrus. The IPL would therefore integrate higher-order functions that require the manipulation of temporal and spatial information[20], which are relevant for sensorimotor interactions with the environment and also for functions such as memory and language. These higher-order functions require the involvement of several cognitive processes mainly supported by posterior parietal regions, such as phonological and semantic processes[36], mental imagery[37], event representation and perceptual attention for memory retrieval (see the model in ref. 25).

The posterior parietal cortex, including the IPS, is traditionally associated with visuospatial and sensorimotor tasks[22,38,39], as well as with numerical cognition[24]. However, the behavioural data derived from the Human Connectome Project do not include such tasks that may also be related to the IPS morphology (see ref. 18 for available evidence on numerical skills). Future studies should investigate the possible associations between sensorimotor/visuospatial abilities and the morphology of the horizontal part of the IPS (hIPS), together with its anterior part, given its direct involvement in these functions and its connections with post-central areas[20].

Although language and memory (especially the verbal one) are typically lateralised to the left hemisphere, we found a strong association with the right hIPS sulcal pattern. It is important to mention that in recent years several researchers have re-evaluated the role of the right, non-dominant, hemisphere in these lateralised cognitive functions[40–43]. For example, in a tractography study, Catani et al.[40] suggested that the relationship between anatomical and functional lateralisation could be further investigated, as they found that a bilateral fractional anisotropy of the perisylvian tracts is associated with better language performance. Furthermore, language and memory are highly susceptible to change with age and learning, and these two functions require a constant balance between neural specialisation, which is likely to be located in the dominant hemisphere, and the neural capacity for functional reorganisation and plasticity, which could recruit the non-dominant hemisphere. Given the above-reported observations of a greater presence of slight deformations of the hIPS fundus (plis de passage) in the left respect to the right hemisphere[28], we might also hypothesise that such fine sulcal morphology could help investigating the process of left lateralisation/functional specialisation. This could explain, at least in part, the lack of relationship

between left hIPS morphology and behaviour in the present study, which focused on gross rather than fine hIPS morphology. These issues should be considered in future investigations by comparing gross vs. fine sulcal morphology and their relationship with behaviour.

Sulcal patterns are trait markers of the brain anatomy that are determined in utero and remain stable throughout life. We therefore speculate that the sulcal pattern in the non-dominant hemisphere may be involved in language and memory functions before their specialisation and lateralisation in the left hemisphere. This argument is consistent with previous longitudinal studies. For example, Budisavljevic et al.[44] observed that the temporo-parietal connections are right-lateralised in early childhood and become progressively more bilateral in adolescence. Furthermore, Brauer and Friederici[45] found that the neural networks supporting language processing are not yet fully specialised in children, also in terms of lateralisation. These findings are in line with the notion that the sulcal pattern may be an early neural fingerprint that constrains initial cognitive development (for a review, see ref. 18). In this framework, we speculate that an interrupted right IPS might allow a better initial development of language and memory functions, providing a neural substrate/reserve for later learning-related plasticity. A similar interpretation has been proposed for reading ability[17]. Following this hypothesis, it would be interesting to further investigate the role of these qualitative features in language and memory functions in children, during the first phases of learning.

Another aspect that deserves further investigation is the relationship between the hIPS morphology and the underlying white matter pattern, as the very embedded position of the IPS raises controversies about its underlying white matter connections. Catani et al.[20] reconstructed the main intra-lobar parietal tracts, identifying a complex set of fibers that could account for the multimodal role of the IPS and explain the morphological variability we found in this study. According to the tension-based hypothesis[46], cortical folding depends in part on the mechanical tension of the fiber tracts that cause the cortex to fold during development. Different sulcal patterns could reasonably result from different connection patterns. In addition, pathways connecting nearby areas should strongly influence the folding patterns, as interareal connections tend to be strongest between nearby areas[7]. For all these reasons, the sulcal morphological patterns observed in the present study may be related to the intra-lobar parietal pathways. In particular, we propose to deepen our knowledge of the Parietal Angular-to-Supramarginal tract (PAS), which connects the two gyri of the inferior parietal lobule (AG and SMG), and the Parietal Inferior to Postcentral tract (PIP), especially the PIP-AG, which connects the post-central gyrus with the angular gyrus and seems to be exquisitely human, as it has not been identified in monkeys. A detailed examination of the white matter could also help us to explain the observed sex-related differences in hIPS morphology, which have not been considered[14,27,28] or found significant[11,16,18] in previous works.

In conclusion, this is the first study to investigate the hIPS sulcal pattern in relation to different cognitive domains in a large sample of healthy participants. Consistent with previous reports in smaller samples, our study supports a more frequent hIPS interruption in the right compared to the left hemisphere. Crucially, the present study showed a robust association between the right hIPS interruption and behavioural performance in the cognitive domains of memory and language. This finding suggests that, despite the leftward lateralisation of these functions, the morphology of the non-dominant hemisphere should be considered, due to its possible mediating role in the early and plastic phases of language and memory development.

## Methods
### Participants
Participants were extracted from the 1200 Subjects Release (S1200) of the Human Connectome Project (HPC) dataset, with the following inclusion criteria: age range: 26–30 years old; MRI field: 3T, DTI data (not analysed for this study). Participants with quality control issues (see https://humanconnectome.org/storage/app/media/documentation/s1200/HCP_

**Table 2 | Cognitive domains, sub-domains (tasks) and metrics from the Human Connectome Project (HCP) database**

| Domain | Sub-domain (Task name) | Metrics |
|---|---|---|
| Memory | Episodic memory (Picture Sequence Memory) | Accuracy |
| | Verbal episodic memory (Penn Word Memory Test) | Accuracy |
| | | Reaction time |
| | Working memory (List Sorting) | Accuracy |
| Executive functions | Cognitive flexibility (Dimensional Change Card Sort) | Accuracy + Reaction time |
| | Inhibition (Flanker Inhibitory Control and Attention Task) | Accuracy + Reaction time |
| Language | Reading decoding (Oral Reading Recognition) | Accuracy |
| | Vocabulary comprehension (Picture Vocabulary) | Accuracy |
| Intelligence | Fluid intelligence (Penn Progressive Matrices) | Accuracy |
| | | Reaction time |
| Self-regulation/ Impulsivity | 200$ reward (Delay Discounting) | AUC (Area Under Curve) |
| | 40 K$ reward (Delay Discounting) | AUC (Area Under Curve) |
| Spatial orientation | Spatial orientation (Variable Short Penn Line Orientation Test) | Accuracy |
| | | Reaction time |
| Attention | Sustained attention (Short Penn Continuous Performance Test) | D' |
| | | Specificity |
| | | Reaction time |
| Speed processing | Processing speed (Pattern Comparison Processing Speed) | Accuracy |

S1200_Release_Reference_Manual.pdf), major neurological diseases, psychiatric or medical disorders were excluded. A total of 390 participants (173 males, 44%) were included in the study.

Participants gave informed consent, and all recruitment and acquisition methods were approved by the Washington University Institutional Review Board (IRB), following all relevant guidelines and regulations.

### Cognitive measures
The HCP database includes various cognitive measures derived from the NIH Toolbox Assessment of Neurological and Behavioural Function (http://www.nihtoolbox.org/), as well as several additional measures assessing cognitive domains not covered by the NIH Toolbox[47]. We used a total of 18 behavioural measures/metrics derived from 12 different cognitive tasks within 8 general cognitive domains (see details in Table 2). Given the large number of cognitive measures and their potential correlation, we used a data reduction approach by performing a principal component analysis (PCA) with oblimin rotation on the scores of all tasks. Notably, this approach allows different tasks to be clustered both within and across different cognitive domains.

### MRI acquisition and pre-processing
The 3T anatomical MRI included in the HPC dataset corresponded to T1-weighted images acquired on Siemens 3T "Connectome Skyra" scanner using a 3D Magnetization Prepared Rapid Acquisition Gradient Echo (MPRAGE) sequence (TR = 2400 ms; TE = 2.14 ms; TI = 1000 ms; flip angle = 8°; FOV = 224 × 224 mm; voxel size = 0.7 mm isotropic). The MRI were preprocessed through the standard FreeSurfer pipeline used in HCP (see full description of these pipelines in ref. 9). T1 MRI were volumetrically registered to MNI152 space using non-linear FNIRT followed by surface registration to Conte69 '164k_fs_LR' mesh[48], with FreeSurfer fsaverage as an intermediate. In order to directly control for potential differences in sulcal morphology leading to the presence of a sulcal interruption induced by the

non-linear spatial normalisation, for a subgroup of 40 randomly selected subjects (20 with interrupted hIPS and 20 with continuous hIPS) we compared the sulcal labelling carried out on raw (non-normalised) and normalised images. The inter-rater reliability calculated between these images was 95% for the right hemisphere and 97.5% for the left hemisphere, thus demonstrating that the normalisation procedure did not affect the presence of sulci interruptions.

Based on the MRI scans of each participant, a 3D model of the cortical surface was then constructed using FreeSurfer. This 3D model included the segmentation of the white matter, the tessellation of the grey/white matter boundary, the inflation of the folded, tessellated surface, and the correction of topological defects. Cortical thicknesses (CT) were calculated from this cortical surface reconstruction by estimating and then refining the grey/white boundary, deforming the surface out to the pial surface, and measuring the distances from each point on the white matter surface to the pial surface[49]. CT was calculated as the closest distance from the grey/white boundary to the grey/CSF boundary at each vertex on the tessellated surface[49].

### Characterisation of the hIPS sulcal pattern
The Connectome Workbench visualisation software was used to superimpose the sulcal maps on the corresponding highly inflated and pial surfaces (as refs. 50,51), by representing the gyral and the sulcal surfaces in light and dark grey, respectively. The individual IPS in the left and right hemispheres were identified from the individual hemispheric aparc.a2009s map of each participant.

The horizontal branch of the IPS (hIPS, a label used in the functional neuroimaging literature, for example refs. 23,24, and by ref. 52, in his three-dimensional sectional anatomy and MRI of the human brain) was then classified according to anatomical criteria. Following the standard atlases of the human cerebral sulci[27,28], we used the post-central sulcus as a landmark to identify the anterior starting point of the hIPS, which extends posteriorly towards the occipital lobe. According to Ono et al.[27], we included in the IPS definition the sulcus that Zlatkina and Petrides[28] called IPS-PO (IntraParietal Sulcus-Paroccipital). However, for the labelling of the hIPS patterns, we only considered the interruptions which divide the parietal portion of hIPS into two or more segments. Therefore, we did not consider the disconnection with the post-central sulcus anteriorly and the short occipital portion of the IPS posteriorly. The boundary between the parietal and occipital portions of the hIPS was derived by extending the dorsal end of the parieto-occipital fissure towards the third branch of the superior temporal sulcus. We considered interrupted any horizontal segment of IPS (dark grey) with a gyral flap (light grey) dividing the sulcus into two or more segments. Therefore, the sulcal pattern of the hIPS was classified as "interrupted" if the course of hIPS showed one or more interruptions by a cortical gyrus, and "continuous" if no interruptions were detected (see Fig. 1).

Visual inspection of the hIPS in the left and right hemispheres was performed blind to potentially confounding information, including participant's age and cognitive scores. The hIPS sulcal patterns were independently classified by two experts (FS, EG; inter-rater reliability of 87% for the left hemisphere and 88% for the right hemisphere). If the two experts disagreed on the classification of a given participant, a third expert (GC) joined the other experts to reach a consensus.

### Statistical analyses
Once the hIPS pattern was characterised in each participant, an exact binomial test with exact Clopper-Pearson 95% CI was performed to investigate the difference in the occurrence of interrupted and continuous morphological patterns in the left and the right hemispheres. Logistic regression was then performed to assess the effect of sex and hemisphere on the likelihood of participants having an interrupted/continuous hIPS. We entered the interruption (yes vs. no) as a dependent variable and both sex (male vs female) and hemisphere (left vs. right) as independent variables. In addition, $\chi^2$ analysis was performed to compare the distribution of our

sample with that of three previous studies investigating the sulcal pattern of the IPS[18,27,28].

A Generalised Linear Model (GLM) was then used to test whether the sulcal pattern of the left and right hIPS was associated with cognitive performance. We included the left hIPS sulcal pattern (interrupted vs continuous), the right hIPS sulcal pattern (interrupted vs continuous) and sex (male vs female) as categorical factors and the different PCA factors as dependent variables. Sex was included in the analysis because of its potential effect on cortical morphology[53,54].

Finally, to control for the possibility that the observed relationship between sulcal pattern and cognitive performance was explained by quantitative measures of morphometry, such as cortical thickness, we performed a targeted analysis on the significant dependent variables resulting from the previous analysis, adding the thickness of the left and right SPL and the left and right IPL as continuous confounding variables in the GLM. All statistical analyses were performed with IBM SPSS Statistic 25.

## Reporting summary

Further information on research design is available in the Nature Portfolio Reporting Summary linked to this article.

## Data availability

No experimental datasets were collected in this study. The MRI and behavioural datasets are freely available from the Human Connectome Project (HPC) website, 1200 Subjects Release (S1200) at this address: https://db.humanconnectome.org/app/template/Login.vm. Selection of the sample used is based on the inclusion and exclusion criteria described in the 'Participants' subsection of the 'Methods' section and can be done from the above website address. Source data underlying Fig. 2 are available in Supplementary Data 1, source data underlying Fig. 3 are available in Supplementary Data 2 and source data underlying Fig. 4 are available in Supplementary Data 3.

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

## Acknowledgements
We thank Marco Catani for comments on an earlier version of this paper.

## Author contributions
F.S., A.C., C.S. and G.C. designed the study. F.S., A.F., E.G., M.R. and G.C. characterised the IPS sulcal pattern. F.S., A.C., A.B., C.S. and G.C. analysed and discussed the data. F.S., A.C., A.B., C.S. and G.C. wrote the paper.

## Competing interests
The authors declare no competing interests.
