## [Peer Review File · Communications Biology]

Reviewers' comments:

Reviewer #1 (Remarks to the Author):

In their study “Intraparietal sulcal morphology affects individual differences in language and memory performance,” Santacroce and colleagues build upon prior work by their co-authors (Roell et al., 2021) and show that the sulcal patterning of the intraparietal sulcus relates to memory and language performance in a substantially-sized sample (for anatomical analyses) of 390 participants.

While there are strengths to this manuscript, there are several major and minor issues that should be addressed prior to publication.

Major Comments

1. Pages 4 & 12. While we do not deny the presence of a relationship between the sulcal patterning of the IPS and cognitive performance, however, we do believe major clarifications are needed with regards to the criteria of what a “fractionated IPS” exactly is, given that there is prominent inter-study ambiguity. Zlatkina and Petrides (2014, Proc Biol Sci) defined the IPS-a and IPS-p as a fractionated main body of the IPS (“IPS”), which entailed a separated vertical component (IPS-a) from a horizontal component (IPS-p). They further classified this IPS as separate from the paroccipital IPS (IPS-PO), which is a more posterior, yet equally as large sulcus (see figure 2 from Zlatkina and Petrides, 2014). The present work and prior work by the authors (Roell et al., 2021 Dev Cogn Neurosci) appear to define a “fractionated hIPS” as a separation between what Zlatkina and Petrides (2014) called the IPS and the IPS-PO (figure 1). The classification by Ono et al. (1990) appears to be the same as the present and prior (Roell et al., 2021) work, that is, a separation between the IPS and IPS-PO (see Figure 9.6 on page 67). These specific anatomical criteria should be clarified in the main text and all figures with brain images (especially figure 1) should be edited accordingly to denote the specific IPS features being discussed.

2. Pages 5 and 9. The comparison of the work’s incidence of IPS types to the literature is particularly useful; however, it may be worth controlling for sample size as each of these studies differs (e.g., Ono et al. (1990) had only 25 brains) and brain image (i.e., post-mortem versus cortical reconstruction). Further, as discussed in the above point, it is also worth noting that the methodologies differ between groups which could also explain the differences in incidence observed.

Minor Comments

1. Page 1. “Affects” infers causality, as this effect is a correlation, using “relates to” would be more applicable.*

2. Page 3. In the second paragraph of the introduction, the authors largely cite their own work on sulcal patterning/incidence and should cite additional studies by other groups and other regions (if applicable).

3. Page 3. The introduction is well-written; however, it may benefit the introduction to include a brief

discussion of what was found as this would be useful to prime readers for the results.

4. Page 5. Considering that the authors conduct analyses relating sex to the presence of a sulcal interruption, including the demographic percentages at the beginning of the results would be useful, rather than only in the methods at the end of the manuscript.

Reviewer #2 (Remarks to the Author):

Santacroce and colleague investigated the sulco-gyral pattern of the left and right horizontal branch of the intraparietal sulcus (hIPS) of 390 healthy participants from the HCP database. They focus on the presence or absence of a sulcus interruption and its relationship with twelve cognitive tasks using principal component analysis and GLM.

They found a relationship between the presence of a right-hIPS interrupted pattern and scores of memory and language performances that they interpret as a morphological advantage.

The study is overall interesting and complements a previous study investigating this morphological signature in relation to numerical cognition. As detailed below, I mainly have concerns regarding the labeling method that might be addressed by providing more details in the method section. My second concern relates to the interpretation of the results and the claim made based on the results.

- It seems that the characterization of the sulci was made on highly inflated and pial surfaces. Was the pattern verified on continuous series of sections (in volumes)? In fact, it is not entirely clear to me how the authors defined an interruption of the sulcus. Was it based on a minimal distance between the two segments that can be quantified? Did the authors check that this continuous versus interrupted pattern does not correspond to *pli de passage* (that might be visible in continuous series of sections in cases where it is buried deep in the depth of a sulcus)? I understand that this morphological pattern (continuous vs interrupted) has been previously described and the authors nicely compared their work to these previous descriptions. Yet, they report a discrepancy (especially in the left hemisphere) and it is important to understand whether the differences in the characterization method between studies might at least in part explain such discrepancy.

- Could the authors provide some details about the preprocessing of the MRI images rather than citing previous papers? In particular, please indicate if the data were normalized to a common space as this might affect the relationship between sulci and gyri (hence the presence of an interruption of sulci). If so, the normalization used (linear or non-linear) and potentially the number of parameters used should also be reported.

- Also, how were the measures of cortical thickness extracted? please provide more details on that aspect as well.

- How were the cognitive measures selected? Was there any rationale? It would have been interesting to include task probing numerical cognition to relate to the previous study from the same group investigating the relationship between these cognitive abilities and hIPS morphological pattern in addition to the other measures. Could the authors comment and describe the strategy behind the choices made here?

- Based on their results, the authors make the claim (page 8) that “The present study reports for the first time the effects of the sulcal pattern of the horizontal intraparietal sulcus (hIPS) on several cognitive functions in a large sample of healthy young adults” and (page 10) they talk about “the cognitive effects of the hIPS sulcal pattern ...” (also at the end of the introduction “an interruption of the hIPS would be functionally advantageous for cognitive functions other than numerical cognition”)

I think that the authors need to tone down their conclusions. While they indeed provide a relationship between a behavioral score and the sulcal pattern of the hIPS, they do not provide a causal relationship, not any relationship between morphological pattern and functional specialization (in terms of brain activity) but instead behavioral measures that might indeed relate to this peculiar morphological sulcal pattern. Relating anatomical pattern with functional specialization is crucial to understand the structure-function relationships. I do think that the study of these morphological pattern is informative and might provide meaningful information about brain function. Yet, at this stage of knowledge, one might be careful about making too strong claim based on a relationship between behavior and brain morphology. Several other morphological characters are present (in terms of number of segments for instance, distance between the segments, relative position of the sulci and segments, etc....) that has not been investigated here and might also contribute to the explain some of the relationship. As acknowledged by the authors, while language and memory are typically lateralized to the left hemisphere, the association between the right hIPS and these functions found in the present study might be studied in relation to the non-dominant hemisphere.

- How do the author interpret the difference between interrupted versus continuous pattern. Could this be due to genetic factors or else ? (see e.g. P. Rakic, Specification of cerebral cortical areas. Science 241, 170–176 (1988). Relatedly, the authors also report differences between males and females in terms of the presence of interrupted hIPS. This result is not discussed. How do the authors interpret this result?

Reviewer #1 (Remarks to the Author):

In their study “Intraparietal sulcal morphology affects individual differences in language and memory performance,” Santacrose and colleagues build upon prior work by their co-authors (Roell et al., 2021) and show that the sulcal patterning of the intraparietal sulcus relates to memory and language performance in a substantially-sized sample (for anatomical analyses) of 390 participants.

While there are strengths to this manuscript, there are several major and minor issues that should be addressed prior to publication.

Major Comments

1) Pages 4 & 12. While we do not deny the presence of a relationship between the sulcal patterning of the IPS and cognitive performance, however, we do believe major clarifications are needed with regards to the criteria of what a “fractionated IPS” exactly is, given that there is prominent inter-study ambiguity. Zlatkina and Petrides (2014, Proc Biol Sci) defined the IPS-a and IPS-p as a fractionated main body of the IPS (“IPS”), which entailed a separated vertical component (IPS-a) from a horizontal component (IPS-p). They further classified this IPS as separate from the paroccipital IPS (IPS-PO), which is a more posterior, yet equally as large sulcus (see figure 2 from Zlatkina and Petrides, 2014). The present work and prior work by the authors (Roell et al., 2021 Dev Cogn Neurosci) appear to define a “fractionated hIPS” as a separation between what Zlatkina and Petrides (2014) called the IPS and the IPS-PO (figure 1). The classification by Ono et al. (1990) appears to be the same as the present and prior (Roell et al., 2021) work, that is, a separation between the IPS and IPS-PO (see Figure 9.6 on page 67). These specific anatomical criteria should be clarified in the main text and all figures with brain images (especially figure 1) should be edited accordingly to denote the specific IPS features being discussed.

Reply: We thank the Reviewer for his/her positive general comment on our manuscript. We now better clarify the anatomical criteria that were used to label the IPS as “continuous” or “interrupted”.

In our study, we focused on the horizontal (parietal) segment of the IPS (hIPS), that is called horizontal in the functional neuroimaging literature (e.g., Dehaene et. al., 2003; Galati et al. 2011) as well as in the anatomical book by Duvernoy (1991). It is characterized as horizontal also by Zlatkina & Petrides (2014; see legend of Figure 1) notwithstanding their use of a different label (i.e. IPS-p).

We identified the IPS according to anatomical criteria, following the atlas of the human cerebral sulci by Ono et al. (1990), that also includes what Zlatkina and Petrides called IPS-PO (or IntraParietal Sulcus-Paroccipital). Therefore, as the Reviewer rightly suggested, our “interrupted pattern” according to Zlatkina and Petrides nomenclature would correspond to the disconnection between IPS and IPS-PO.

Moreover, according to Ono’s nomenclature, the posterior end of the IPS (i.e. the IPS-PO) almost always extends into the occipital lobe, thus including a short occipital segment (see figure 9.9, page 70 by Ono et al., 1990). Given our declared interest on the parietal lobe, we excluded this occipital segment from our labelling and analysis. The boundary between the parietal and occipital portions of the hIPS was derived by extending the dorsal end of the parietoccipital fissure towards the third branch of the superior temporal sulcus. According to this procedure, we labelled as “interrupted” any horizontal IPS (without the occipital segment of Ono’s IPS and of Zlatkina & Petrides’ IPS-PO) with an interruption that breaks it into two or more segments falling in the parietal lobe.

We now included these specifications in the revised version of the manuscript.

Revised Text:

Methods (paragraph 4.4, page 12)

*The Connectome Workbench visualization software was used to superimpose the sulcal maps on the corresponding highly inflated and pial surfaces (as Voorhies et al. 2021; Yao et al. 2023), by representing the gyral and the sulcal surfaces in light and dark grey, respectively. The individual IPS in the left and right hemispheres were identified from the individual hemispheric *aparca2009s* map of each participant.*

The horizontal branch of the IPS (hIPS, a label used in the functional neuroimaging literature, see for example Dehaene et al. 2003 and Galati et al. 2011, and by Duvernoy 1991, in his three-dimensional sectional anatomy and MRI of the human brain) was then classified according to anatomical criteria. Following the standard atlases of the human cerebral sulci (Ono et al., 1990 and Zlatkina & Petrides, 2014), we used the post-central sulcus as a landmark to identify the anterior starting point of the hIPS, which extends posteriorly towards the occipital lobe. According to Ono et al. (1990), we included in the IPS definition the sulcus that Zlatkina and Petrides (2014) called IPS-PO (IntraParietal Sulcus-Paroccipital). However, for the labelling of the hIPS patterns, we only considered the interruptions which divide the parietal portion of hIPS into two or more segments. Therefore, we did not consider the disconnection with the postcentral sulcus anteriorly and the short occipital portion of the IPS posteriorly. The boundary between the parietal and occipital portions of the hIPS was derived by extending the dorsal end of the parieto-occipital fissure towards the third branch of the superior temporal sulcus. We considered interrupted any horizontal segment of IPS (dark gray) with a gyral flap (light grey) dividing the sulcus into two or more segments. Therefore, the sulcal pattern of the hIPS was classified as "Interrupted" if the course of hIPS showed one or more interruptions by a cortical gyrus, and "Continuous" if no interruptions were detected (see Figure 1).

2) Pages 5 and 9. The comparison of the work's incidence of IPS types to the literature is particularly useful; however, it may be worth controlling for sample size as each of these studies differs (e.g., Ono et al. (1990) had only 25 brains) and brain image (i.e., post-mortem versus cortical reconstruction). Further, as discussed in the above point, it is also worth noting that the methodologies differ between groups which could also explain the differences in incidence observed.

Reply: Following the reviewer comments, methodological differences across studies (regarding sample size, type of images and labelling procedure) were better acknowledged. The sample size was already taken into consideration in the original version of the manuscript (refer to the χ^2 analysis) but this issue was not highlighted. We now explicitly discuss this point in the Results section (page 5). As we could not statistically control for image type, acquisition parameters and labelling method, (page 9) we now mention these differences in the revised version of the Discussion. In addition, we also discuss the potential differences across studies in the detection of interruptions as they might explain some discrepancies, especially concerning the left hemisphere (see also our Reply to Point 1 of Reviewer #2). Importantly, this issue likely does not affect our main finding regarding the relationship between the morphology of the right IPS and behaviour, since the frequency of right morphological patterns is the same across studies.

Revised Text:

Results (page 5)

We then compared the distribution of the hIPS sulcal pattern observed in the present study with that of the three previous studies that investigated the hIPS sulcal morphology (Ono et al., 1990; Zlatkina & Petrides, 2014; Roell et al., 2021). The χ^2 analysis, controlling for the sample size, showed a significant difference between the four studies regarding the distribution of the left IPS sulcal pattern ($\chi^2= 14.1$; $df=3$; $p=0.003$).

Discussion (page 9)

In this regard, it is important to note the large sample size of the current study ($N=390$) compared to previous studies. Ono et al. (1990) examined only 25 post-mortem brains, Zlatkina & Petrides (2014) analysed MRI from 40 participants and Roell et al. (2021) examined the effect of the hIPS interruption in 98 participants. Overall, these studies differ for several methodological aspects (sample size, brain images, interruption detection, labelling criteria) and this might have influenced the inconsistent morphological pattern distributions in the left hemisphere. In particular, the study which reported a significantly higher proportion of interruptions in the left hemisphere (Zlatkina & Petrides, 2014; see Figure 3) is the only one considering the "plis de passage" (also known as submerged gyri or gyral/cortical bridges, see Mangin et al., 2019) as sulcal interruptions, and finding them as more frequent in the left hemisphere (Zlatkina & Petrides, 2014).

Minor Comments

1) Page 1. "Affects" infers causality, as this effect is a correlation, using "relates to" would be more applicable.*

Reply: As suggested by the Reviewer, we changed the title into: "INTRAPARIETAL SULCAL MORPHOLOGY RELATES TO INDIVIDUAL DIFFERENCES IN LANGUAGE AND MEMORY PERFORMANCE".

2) Page 3. In the second paragraph of the introduction, the authors largely cite their own work on sulcal patterning/incidence and should cite additional studies by other groups and other regions (if applicable).

Reply: We agree with the Reviewer on this point. We added the following references about the relationship between morphology and cognition/behaviour:

- Fedeli D, Del Maschio N, Caprioglio C, Sulpizio S, Abutalebi J. Sulcal Pattern Variability and Dorsal Anterior Cingulate Cortex Functional Connectivity Across Adult Age. *Brain Connect.* 2020 Aug;10(6):267-278. doi: 10.1089/brain.2020.0751. Epub 2020 Jul 20. PMID: 32567343.
- Nakamura M, Nestor PG, McCarley RW, Levitt JJ, Hsu L, Kawashima T, Niznikiewicz M, Shenton ME. Altered orbitofrontal sulcogyral pattern in schizophrenia. *Brain.* 2007 Mar;130(Pt 3):693-707. doi: 10.1093/brain/awm007. PMID: 17347256; PMCID: PMC2768130.

3) Page 3. The introduction is well-written; however, it may benefit the introduction to include a brief discussion of what was found as this would be useful to prime readers for the results.

Reply: As recommended, we added a short overview of the main results of our study at the end of the Introduction (page 4).

Revised text:

Introduction (page 4)

Finally, we analysed the association between hIPS morphology and the PCA cognitive components with the hypothesis that an interruption of the hIPS would be functionally advantageous for cognitive functions in addition to numerical cognition as previously described (Roell et. al., 2021).

Our findings revealed a prevalence of the "interrupted" pattern in the right hemisphere, which is associated with higher behavioral performance in memory and language tasks, putatively due to intra-lobar parietal white matter.

4) Page 5. Considering that the authors conduct analyses relating sex to the presence of a sulcal interruption, including the demographic percentages at the beginning of the results would be useful, rather than only in the methods at the end of the manuscript.

Reply: As suggested, we added demographic percentage of the sample at the beginning of the Results section.

Revised text:

Results (page 4)

Our sample consisted of 390 participants (173 males, 40% - age range 26-30) derived from the Human Connectome Project (HCP).

Reviewer #2 (Remarks to the Author):

Santacrose and colleague investigated the sulco-gyral pattern of the left and right horizontal branch of the intraparietal sulcus (hIPS) of 390 healthy participants from the HCP database. They focus on the presence or absence of a sulcus interruption and its relationship with twelve cognitive tasks using principal component analysis and GLM.

They found a relationship between the presence of a right-hIPS interrupted pattern and scores of memory and language performances that they interpret as a morphological advantage.

The study is overall interesting and complement a previous study investigating this morphological signature in relation to numerical cognition. As detailed below, I mainly have concerns regarding the labeling method that might be addressed by providing more details in the method section. My second concern relates to the interpretation on the results and the claim made based on the results.

1) It seems that the characterization of the sulci was made on highly inflated and pial surfaces. Was the pattern verified on continuous series of sections (in volumes)? In fact, it is not entirely clear to me how the authors defined an interruption of the sulcus. Was it based on a minimal distance between the two segments that can be quantified? Did the authors check that this continuous versus interrupted patterns do not correspond to pli de passage (that might be visible continuous series of sections in cases whether it is buried deep in the depth of a sulcus)? I understand that this morphological pattern (continuous vs interrupted) has been previously described and the authors nicely compared their work to these previous descriptions. Yet, they report discrepancy (especially in the left hemisphere) and it is important to understand whether the differences in the characterization method between studies might at least in part explain such discrepancy.

Reply: We are very grateful to the Reviewer for the thoroughness with which he/she has read our work. We find his/her comments extremely useful for improving the article and for further studies.

As the Reviewer pointed out, we characterized the sulci on highly inflated and pial surfaces, like other authors have recently done for different sulci by using FreeSurfer software (Miller et al. 2021, Voorhies et al. 2021, Yao et al. 2023). This 3D approach is relatively similar to the one based on 3D brain reconstructions by Brainvisa software (e.g. Cachia et al. 2018, Roell et al. 2021). As in all these previous 3D studies, we did not characterize or confirm the sulcal patterns on continuous series of 2D volume slices. Instead, we superimposed the sulcal maps of each subject on its highly inflated image by using the Connectome Workbench standard settings (sulcal depth colour parameters <0 with 0 = mid depth). Using this approach, the light grey color represents the gyral surface (from mid depth to surface) and the dark grey color represents the sulcal surface (from mid depth to max depth). We considered interruption as any horizontal segment of IPS that has a gyral flap (light grey) dividing it into two or more segments, independently of the length of the flap and of the distance between segments. Therefore, we did not consider the interruption by a 'pli de passage' if the latter was deeper than mid depth. In other words, we evaluated the gross morphology disregarding the slight deformation of the hIPS fundus.

We have added these specifications in the revised version of the manuscript (Methods section 4.4, page 12).

We also now discuss in greater detail the methodological differences in sulcus characterization across the available studies on IPS, and their possible role on the observed discrepancies concerning the left IPS. Zlatkina and Petrides (2014) explicitly included the interruptions represented by plis de passage in their classification of IPS, using continuous series of sections for their labelling, while Ono et. al. (1990) distinguished "true" from "pseudo" (i.e. plis de passage) interruptions but did not consider this distinction in their classification. Interestingly, Zlatkina and Petrides (2014, page 3) found that interruptions by a pli de passage represent the great majority (84,4%) of the total interruptions in the left hemisphere, while they are less frequent (63,6%) in the right hemisphere. It is therefore possible that at least part of the observed discrepancy across studies concerning the left IPS morphology is explained by the differences in the method used for the detection of sulcal interruption. Notably, the only study which explicitly included plis de passage (Zlatkina & Petrides, 2014) for the definition of an IPS interruption is the study describing the highest proportion of interruptions in the left hemisphere, where plis de passage appeared to be more frequent.

Based on these observations, we could hypothesize that the use of fine sulcal morphological features like the plis de passage could help investigating the process of lateralisation/functional specialization in the dominant (left) hemisphere. This could explain, at least in part, the lack of relationship between the left hIPS morphology and behaviour in the present study, which focused on gross rather than fine hIPS morphology. Future investigations might compare gross vs. fine sulcal morphology and their relationship with behaviour. We now added these considerations in the revised version of the manuscript and thank again the Reviewer for having contributed to what we consider to be an important improvement of the manuscript.

Revised text:

Discussion (page 9)

In this regard, it is important to note the large sample size of the current study (N=390) compared to previous studies. Ono et. al. (1990) examined only 25 post-mortem brains, Zlatkina & Petrides (2014) analysed MRI from 40 participants and Roell et. al. (2021) examined the effect of the hIPS interruption in 98 participants. Overall, these studies differ for several methodological aspects (sample size, brain images, interruption detection, labelling criteria) and this might have influenced the inconsistent morphological pattern distributions in the left hemisphere. In particular, the study which reported a significantly higher proportion of interruptions in the left hemisphere (Zlatkina & Petrides, 2014; see Figure 3) is the only one considering the “plis de passage” (also known as submerged gyri or gyral/cortical bridges, see Mangin et al., 2019) as sulcal interruptions, and finding them as more frequent in the left hemisphere (Zlatkina & Petrides, 2014).

Discussion (page 10)

Furthermore, language and memory are highly susceptible to change with age and learning and these two functions require a constant balance between neural specialisation, which is likely to be located in the dominant hemisphere, and the neural capacity for functional reorganisation and plasticity, which could recruit the non-dominant hemisphere. Given the above reported observations of a greater presence of slight deformations of the hIPS fundus (plis de passage) in the left respect to the right hemisphere (Zlatkina & Petrides, 2014), we might also hypothesize that such fine sulcal morphology could help investigating the process of left lateralisation/functional specialization. This could explain, at least in part, the lack of relationship between left hIPS morphology and behaviour in the present study, which focused on gross rather than fine hIPS morphology. These issues should be considered in future investigations by comparing gross vs. fine sulcal morphology and their relationship with behaviour.

Methods (paragraph 4.4, page 12)

*The Connectome Workbench visualization software was used to superimpose the sulcal maps on the corresponding highly inflated and pial surfaces (as Voorhies et al. 2021; Yao et al. 2023), by representing the gyral and the sulcal surfaces in light and dark grey, respectively. The individual IPS in the left and right hemispheres were identified from the individual hemispheric *aparc.a2009s* map of each participant.*

The horizontal branch of the IPS (hIPS, a label used in the functional neuroimaging literature, see for example Dehaene et. al. 2003 and Galati et al. 2011, and by Duvernoy 1991, in his three-dimensional sectional anatomy and MRI of the human brain) was then classified according to anatomical criteria. Following the standard atlases of the human cerebral sulci (Ono et al., 1990 and Zlatkina & Petrides, 2014), we used the post-central sulcus as a landmark to identify the anterior starting point of the hIPS, which extends posteriorly towards the occipital lobe. According to Ono et. al. (1990), we included in the IPS definition the sulcus that Zlatkina and Petrides (2014) called IPS-PO (IntraParietal Sulcus-Paroccipital). However, for the labelling of the hIPS patterns, we only considered the interruptions which divide the parietal portion of hIPS into two or more segments. Therefore, we did not consider the disconnection with the postcentral sulcus anteriorly and the short occipital portion of the IPS posteriorly. The boundary between the parietal and occipital portions of the hIPS was derived by extending the dorsal end of the parieto-occipital fissure towards the third branch of the superior temporal sulcus. We considered interrupted any horizontal segment of IPS (dark gray) with a gyral flap (light grey) dividing the sulcus into two or more segments. Therefore, the sulcal pattern of the hIPS was classified as “Interrupted” if the course of hIPS showed one or more interruptions by a cortical gyrus, and “Continuous” if no interruptions were detected (see Figure 1).

2) Could the authors provide some details about the preprocessing of the MRI images rather than citing previous papers? In particular, please indicate if the data were normalized to a common space as this might affect the relationship between sulci and gyri (hence the presence of an interruption of sulci). If so, the normalization used (linear or non-linear) and potentially the number of parameters used should also be reported.

Reply: The MRI were preprocessed using the standard FreeSurfer pipeline used in HCP. T1 MRI were volumetrically registered to MNI152 space using nonlinear FNIRT followed by surface registration to

Conte69 '164k_fs_LR' mesh (Van Essen et al. 2012), with FreeSurfer fsaverage as an intermediate. In order to directly test for potential differences in sulcal morphology leading to the presence of a sulcal interruption induced by the non-linear spatial normalization, we performed an additional analysis on a subset of 40 randomly-selected participants (20 with interrupted hIPS and 20 with continuous hIPS) comparing the sulcal labelling carried out on raw (non-normalized) images and on normalized images. As detailed in the table below, the spatial normalisation does not impact the sulcal labelling process. Indeed, the inter-rater reliability calculated between the normalized and non-normalized images of this subgroup is 95% for the right hemisphere and 97,5% for the left hemisphere.

Left hemisphere:

	Interrupted hIPS (with normalization)	Continuous hIPS (with normalization)
Interrupted hIPS (without normalization, raw data)	20 (50%)	1 (2,5%)
Continuous hIPS (without normalization, raw data)	0 (0%)	19 (47,5%)

Right hemisphere:

	Interrupted hIPS (with normalization)	Continuous hIPS (with normalization)
Interrupted hIPS (without normalization, raw data)	20 (50%)	2 (5%)
Continuous hIPS (without normalization, raw data)	0 (X%)	18 (45%)

We now added this specification and the new analysis in the revised version of the manuscript.

Revised text:

Methods (paragraph 4.3, page 12)

The MRI were preprocessed through the standard FreeSurfer pipeline used in HCP (see full description of these pipelines in Glasser et al. 2013). T1 MRI were volumetrically registered to MNI152 space using nonlinear FNIRT followed by surface registration to Conte69 '164k_fs_LR' mesh (Van Essen et al. 2012), with FreeSurfer fsaverage as an intermediate. In order to directly control for potential differences in sulcal morphology leading to the presence of a sulcal interruption induced by the non-linear spatial normalization, for a subgroup of 40 randomly selected subjects (20 with interrupted hIPS and 20 with continuous hIPS) we compared the sulcal labelling carried out on raw (non-normalized) and normalized images. The inter-rater reliability calculated between these images was 95% for the right hemisphere and 97,5% for the left hemisphere, thus demonstrating that the normalization procedure did not affect the presence of sulci interruptions.

3) Also, how were the measures of cortical thickness extracted? please provide more details on that aspect as well.

Reply: We extracted the measures of cortical thickness as follows. Based on the MRI scans of each participant, a 3D model of the cortical surface was constructed using FreeSurfer. This 3D model included the segmentation of the white matter, the tessellation of the gray/white matter boundary, the inflation of the folded, tessellated surface, and the correction of topological defects. Cortical thicknesses (CT) were calculated from this cortical surface reconstruction by estimating and then refining the gray/white boundary, deforming the surface out to the pial surface, and measuring the distances from each point on the white matter surface to the pial surface (Fischl & Dale, 2000). CT was calculated as the closest distance from the gray/white boundary to the gray/CSF boundary at each vertex on the tessellated surface (Fischl & Dale, 2000).

Revised text:

Methods (paragraph 4.3, page 12)

Based on the MRI scans of each participant, a 3D model of the cortical surface was then constructed using FreeSurfer. This 3D model included the segmentation of the white matter, the tessellation of the gray/white matter boundary, the inflation of the folded, tessellated surface, and the correction of topological defects. Cortical thicknesses (CT) were calculated from this cortical surface reconstruction by estimating and then refining the gray/white boundary, deforming the surface out to the pial surface, and measuring the distances from each point on the white matter surface to the pial surface (Fischl & Dale, 2000). CT was calculated as the closest distance from the gray/white boundary to the gray/CSF boundary at each vertex on the tessellated surface (Fischl & Dale, 2000).

4) How were the cognitive measures selected? Was there any rationale? It would have been interesting to include task probing numerical cognition to relate to the previous study from the same group investigating the relationship between these cognitive abilities and hIPS morphological pattern in addition to the other measures. Could the authors comment and describe the strategy behind the choices made here?

Reply: We selected all the available cognitive tasks (18 measures from 12 tasks) provided by the Human Connectome Project (HCP). Unfortunately, HCP does not provide tasks probing numerical cognition. We agree with the Reviewer that it would have been very interesting to replicate previous group studies investigating the relationship between hIPS sulcation and numerical cognition. This issue is now mentioned in the Discussion section.

Revised text:

Discussion (page 9)

The posterior parietal cortex, including the IPS, is traditionally associated with visuospatial and sensorimotor tasks (Colby et al. 1996; Mesulam 1999; Corbetta & Shulman 2002), as well as with numerical cognition (Dehaene et al., 2003). However, the behavioural data derived from the Human Connectome Project do not include such tasks that may also be related to the IPS morphology (see Roell et al., 2021, for available evidence on numerical skills).

5) Based on their results, the authors make the claim (page 8) that “The present study reports for the first time the effects of the sulcal pattern of the horizontal intraparietal sulcus (hIPS) on several cognitive functions in a large sample of healthy young adults” and (page 10) they talk about “the cognitive effects of the hIPS sulcal pattern ...” (also at the end of the introduction “an interruption of the hIPS would be functionally advantageous for cognitive functions other than numerical cognition”)

I think that the authors need to tone down their conclusions. While they indeed provide a relationship between a behavioral score and the sulcal pattern of the hIPS, they do not provide a causal relationship, not any relationship between morphological pattern and functional specialization (in terms of brain activity) but instead behavioral measures that might indeed relate to this peculiar morphological sulcal pattern. Relating anatomical pattern with functional specialization is crucial to understand the structure-function relationships. I do think that the study of these morphological pattern is informative and might provide meaningful information about brain function. Yet, at this stage of knowledge, one might be careful about making too strong claim based on a relationship between behavior and brain morphology. Several other morphological characters are present (in terms of number of segments for instance, distance between the segments, relative position of the sulci and segments, etc....) that has not been investigated here and might also contribute to the explain some of the relationship. As acknowledged by the authors, while language and memory are typically lateralized to the left hemisphere, the association between the right hIPS and these functions found in the present study might be studied in relation to the non-dominant hemisphere.

Reply: We agree with the Reviewer that other morphological characteristics, not considered in the present work, might have contributed to the observed results. In addition, as the Reviewer rightly pointed out, our

data do not provide a causal relationship between hIPS morphological pattern and cognitive specialization, also given the absence of a functional correlate. We have therefore toned down our conclusions, replacing terms such as “effect/s” and “functionally” with “relation” or “association” in the Abstract, Introduction and Discussion, as well as in the title of Figure 4. We also revised the beginning of the final conclusions as follows: “In conclusion, this is the first study to investigate the hIPS sulcal pattern in relation to different cognitive domains in a large sample of healthy participants.”

6) How do the author interpret the difference between interrupted versus continuous pattern. Could this be due to genetic factors or else ? (see e.g. P. Rakic, Specification of cerebral cortical areas. *Science* 241, 170–176 (1988). Relatedly, the authors also report differences between males and females in terms of the presence of interrupted hIPS. This result is not discussed. How do the authors interpret this result?

Reply: We thank the Reviewer for the possibility of better explaining the mechanisms underlying sulci morphology and the role of sex.

As suggested by the Reviewer, genetic and epigenetic factors were early proposed in the cortical folding debate by Rakic (1988) since the cortical folding process occurs during prenatal brain development, leading researchers to assign a strong importance to the genetic component. Specifically, the most influential factors are the rapid increase in the number of neurons putting pressure on the limited space inside the skull, the combination of radial and tangential forces due to the migration of neurons and their variation in growth rate, respectively, and the variation in density, orientation and connections of white matter fibres (see Cachia et al. 2021 for review). These factors are guided by specific gene expression and the mutation of these genes can lead to disruption of cortical development and folding (see Van Essen, 2020 for a review). The important role of white matter fibres (axonal tracts) suggests that the presence of a sulcus interruption by a gyrus is, at least in part, related to the higher number, greater fibre diameter and/or greater myelination in the bundles connecting near cortical areas. In our case, intra-lobar parietal subcortical connections should be associated to enhanced circumvolution of parietal lobe and interrupted hIPS. As stated at pages 10 and 11 of the original manuscript, we speculate that the Parietal Angular-to-Supramarginal tract (PAS), which connects the two gyri of the inferior parietal lobule (AG and SMG), and the Parietal Inferior to Post-central tract (PIP), which connects the post-central gyrus with the angular gyrus, may be good candidates for the genesis of hIPS interruption. Another not necessarily incompatible hypothesis is a putative increase in IPS surface area. hIPS sulcal interruption might therefore be an indirect marker of an increase in the amount of cortical tissue and connections available for neuroplastic processes underlying the acquisition of cognitive functions. Finally, cytoarchitectonic differences may also contribute to the observed cognitive differences. Indeed, several studies suggest a relationship between cortical folding and the underlying cytoarchitecture (Fischl et al. 2007; Glasser et al. 2016; Unger et al. 2023).

We have now addressed this issue more in detail both in the Introduction and the Discussion, by reporting a reference to the available knowledge on the genetic factors that have been proposed for the genesis of cortical folding.

For what concerns the role of sex on the morphology results, the original version of the manuscript did not explicitly discuss the issue because we were primarily focused on the morphology-cognition relationship, and we did not observe any significant interaction between sex and hIPS pattern related to behavioural performance. However, the Reviewer correctly pointed out that sex was significantly related to the hIPS pattern, with males exhibiting significantly higher presence of interrupted hIPS compared to females. Sex were reported to affect the sulcal quantitative anatomy (Duchesnay et. al., 2007 ; Fish et al. 2017) but hIPS is not included among the sulci that have shown such an effect. Moreover, in the studies on sulci morphology discussed in our manuscript, either sex has not been considered (eg. Ono et. al., 1990; Fornito et. al., 2004; Zlatkina & Petrides, 2014) or no significant sex effect was observed (eg. Del Maschio et. al., 2019; Fedeli et. al., 2020; Roell et. al., 2021). For this reason, we do not have a clear explanation for the observed result, and we believe that further investigations are needed to draw a firm conclusion. However, we can speculate that a sex difference in sulci qualitative morphology might be somehow related to, still unknown, subcortical or genetic feature. We now discuss this speculation before the conclusions, when mentioning the importance of deepening the knowledge on white matter fibers.

Revised text:Introduction (page 3)

Genetic factors have been early proposed in the cortical folding debate (e.g. Rakic, 1988), with particular attention to the rapid increase in the number of neurons putting pressure on the limited space inside the skull, the combination of radial and tangential forces (due to the migration of neurons and their variation in growth rate, respectively), and the variation in density, orientation and connections of white matter fibres (see Van Essen 2020 and Cachia et al., 2021 for reviews). Moreover, a relationship between cortical folding and the underlying cytoarchitecture has been suggested (Fischl et al. 2007; Glasser et al. 2016; Unger et al. 2023)..[....] The authors suggested that the advantage of an interrupted/sectioned pattern might be, at least in part, related to the higher number, greater fibre diameter and/or greater myelination of the underlying white matter.

Discussion (page 9)

Regarding the association between the hIPS sulcal pattern and cognition, we found a strong relation of the right hIPS pattern with the performance in memory and language tasks. Specifically, a right hIPS interruption was associated with higher scores on the first PCA factor, which includes episodic memory (verbal and non-verbal), working memory and language (production and comprehension) tasks. Consistent with previous evidence (Borst et al., 2016; Cachia et. al., 2018; Roell et. al., 2021), sulcal interruption appears to be advantageous and therefore to represent a positive, early defined, fingerprint for the cognitive development. The presence of a sulcal interruption may reflect an increase of the underlying microstructure of both the white matter bundles and the cortical ribbon (Van Essen, 2020). Therefore, it might represent an indirect marker of an increased amount of cortical tissue and connections available for neuroplastic processes underlying the acquisition of cognitive functions.

Discussion (page 10)

In particular, we propose to deepen our knowledge of the Parietal Angular-to-Supramarginal tract (PAS), which connects the two gyri of the inferior parietal lobule (AG and SMG), and the Parietal Inferior to Post-central tract (PIP), especially the PIP-AG, which connects the post-central gyrus with the angular gyrus and seems to be exquisitely human, as it has not been identified in monkeys. A detailed examination of the white matter could also help us to explain the observed sex-related differences in hIPS morphology, which have not been considered (e.g. Ono et. al., 1990; Fornito et. al., 2004; Zlatkina & Petrides, 2014) or found as significant (e.g. Del Maschio et. al., 2019; Fedeli et. al., 2020; Roell et. al., 2021) in previous works.

REVIEWERS' COMMENTS:

Reviewer #1 (Remarks to the Author):

The authors have addressed each of my concerns. My one remaining minor comment is that it would be nice to see individual data points in Figure 4.

Reviewer #2 (Remarks to the Author):

The authors' responses and revisions clarified all my concerns and I have no further comments.

Reviewer #1 (Remarks to the Author):

The authors have addressed each of my concerns. My one remaining minor comment is that it would be nice to see individual data points in Figure 4.

Reply: We thank the Reviewer for the last minor comment that allowed us to improve results description. As suggested, we showed the individual data points and box plots in Figure 4.